# Design Principles of Large Cation Incorporation in Halide Perovskites

**DOI:** 10.3390/molecules26206184

**Published:** 2021-10-13

**Authors:** Heesoo Park, Syam Kumar, Sanjay Chawla, Fedwa El-Mellouhi

**Affiliations:** 1Qatar Environment and Energy Research Institute, Hamad Bin Khalifa University, Doha P.O. Box 34110, Qatar; hpark@hbku.edu.qa; 2Department of Health and Nutritional Science, Institute of Technology Sligo, Ash Lane, F91 YW50 Sligo, Ireland; Kumar.Syam@itsligo.ie; 3Qatar Computing Research Institute, Hamad Bin Khalifa University, Doha P.O. Box 34110, Qatar; schawla@hbku.edu.qa

**Keywords:** halide perovskites, photocatalyst, density functional theory, large cation, fluorinated cation, octahedral deformation, mixed cation, non-covalent interaction, Bayesian optimization

## Abstract

Perovskites have stood out as excellent photoactive materials with high efficiencies and stabilities, achieved via cation mixing techniques. Overcoming challenges to the stabilization of Perovskite solar cells calls for the development of design principles of large cation incorporation in halide perovskite to accelerate the discovery of optimal stable compositions. Large fluorinated organic cations incorporation is an attractive method for enhancing the intrinsic stability of halide perovskites due to their high dipole moment and moisture-resistant nature. However, a fluorinated cation has a larger ionic size than its non-fluorinated counterpart, falling within the upper boundary of the mixed-cation incorporation. Here, we report on the intrinsic stability of mixed Methylammonium (MA) lead halides at different concentrations of large cation incorporation, namely, ehtylammonium (EA; [CH_3_CH_2_NH_3_]^+^) and 2-fluoroethylammonium (FEA; [CH_2_FCH_2_NH_3_]^+^). Density functional theory (DFT) calculations of the enthalpy of the mixing and analysis of the perovskite structural features enable us to narrow down the compositional search domain for EA and FEA cations around concentrations that preserve the perovskite structure while pointing towards the maximal stability. This work paves the way to developing design principles of a large cation mixture guided by data analysis of DFT data. Finally, we present the automated search of the minimum enthalpy of mixing by implementing Bayesian optimization over the compositional search domain. We introduce and validate an automated workflow designed to accelerate the compositional search, enabling researchers to cut down the computational expense and bias to search for optimal compositions.

## 1. Introduction

The majority of the world’s population is moving towards green energy technologies, out of which solar energy is considered the most suitable form to harness. Over the last two decades, the family of materials known as Halide Perovskites have stood out as being excellent photoactive materials, thanks to their exceptional formability and bandgap tunability [1].

The perovskite family of materials offers a huge opportunity for the compositional and structural tuning that opens the possibility of discovering high performance materials for energy conversion. Halide perovskites ($ABX_{3}$) consist of metal atoms or organic molecules at the *A*-site; Ge, Pb, and Sn at the *B*-site; and I, Br, and Cl at the *X*-site [2,3]. Methylammonium lead halide (CH_3_NH_3_PbI_3_) is the mainstay of the perovskite family due to its bandgap of 1.5 eV and its facile fabrication by low-temperature solution processing. The baseline compound for numerous mixed halide perovskites reported in the past few years has been optimized to obtain better stabilities and photoconversion efficiencies [4]. The use of perovskites and their derivatives in the field of photovoltaics has been intensely researched ever since its discovery, and a vast number of articles and reviews are already available, starting from their fundamentals up to photovoltaic module physics [5,6,7].

Several of the materials that belong to this family have shown power conversion efficiencies (PCE) up to 25.5% [8] due to an exceptional combination of properties suitable for photoconversion such as (i) bandgaps, (ii) charge carrier mobilities and diffusion lengths, (iii) lower surface recombination rates, (iv) high optical absorption, (v) structural defect tolerance and (vi) defect characteristics. However, halide perovskites exhibit long-term performance constraints [9,10,11] such as degradation under thermal treatment, anomalous photocurrent with voltage (J-V) hysteresis and susceptibility to hydrolysis. These constraints stand as the bottleneck against the large-scale deployment of these materials in the photovoltaics market [12].

In addition to photovoltaic applications, Sunghak et al. recently demonstrated the splitting of the aqueous HI solution into H_2_ and I_3_^−^ with methylammonium lead halide perovskite under solar irradiation [13]. Halide perovskites have been reported for visible-light water-splitting and organic pollutant degradation such as CO_2_ reduction, which can be collectively classified under photocatalysis [14,15,16,17]. Their suitable bandgap and band edge positions can activate a wide range of redox reactions since they can be tuned via compositional optimization [17]. The relative positions of the conduction band (CB) generally determine the reduction ability for H_2_ generation and CO_2_ reduction; while those of the valence band (VB) are critical for water oxidation, allowing H_2_O splitting. In addition to the above suitable light absorption properties, halide perovskites possess properties that enhance their photocatalytic activity, such as charge migration/stability, and the combination of their ferroelectric and piezoelectric effects [3,5,18].

Despite the perovskites stability issues under visible light irradiation and moisture conditions during photocatalytic device operation [18], there have been recent reports on water-stable halide perovskites that are useful for H_2_ production from direct water splitting. Ju et al. showed a single crystal of [(CH_3_)_2_NH_2_]SnI_3_ is stable in pure water for up to 16 hours and does not dissolve [19]. Pisanu et al. uncovered that the bromide analogue ([(CH_3_)_2_NH_2_]SnBr_3_) has a similar stability while generating H_2_ from deionized water without dissolving [20]. Although this material formed an opalescent solution in the deionized water, the authors could recover the pristine phase perovskite as soon as the material was dried. Alternatively, as proposed by Ding et al., the nanocrystalline (NC) glass formation can improve the stability of the inorganic counterparts such as CsPbBr_3_ [21]. The formation of NC glass solves the rapid materials deactivation in water by fostering the encapsulation effect.

In *A*PbI_3_ perovskites, the contribution to band edge states originates from the PbI_6_ octahedra, where the Pb−I bond lengths and angles determine the orbital overlap that modulates the positions of the CB and VB [4]. For instance, the replacement of an MA cation with a larger FA cation results in the reduction of bandgap by 0.05 eV while the replacement with a smaller Cs ion increases the bandgap by 0.16 eV [3,22]. Hence, *A*-site cations contribute very weakly to the band edges but strongly influence the perovskites phase stability due to their size and dynamics [23,24]. A common approach to remaining within the bandgap range of pristine lead halide perovskites has been the partial incorporation of the cations. Interestingly, there are recent reports that demonstrate a promising route of the commercialization of halide perovskite solar cells light absorbers based on formamidinium (FA) and a mixture of other cations [25,26,27]. In addition to MA and FA, there are other commonly used monovalent cations such as ethylammonium (EA), guanidinium (GUA), hydrazinium (HZ), and hydroxylammonium (HA) [28,29]. Furthermore, sulfonium-based cations, such as trimethylsulfonium and trimethylsulfoxonium, have shown the formation of lower dimensional perovskites resulting in a stability increase, as their ionic size is bulkier than MA [30,31].

A mixture of the above cations were found to positively affect the dynamics of the organic molecule, thus offering the possibility of stabilizing the perovskite structure due to entropic effects and by minimizing the internal energy [32].

Alongside these aliphatic organic cations used in perovskites, Huang et al. reported the successful use of a halogenated-methylammonium cation by synthesizing the halide perovskite, CH_2_FNH_3_PbBr_3_ [33]. Interestingly, the resulting crystal structure of the synthesized perovskite material preserved a 3-dimensional crystal structure (3D) while having a slightly larger bandgap (2.3 eV) than MAPbBr_3_ (2.2 eV) with the enhanced absorption coefficient and smaller exciton binding energies compared to MA-based compounds. Interestingly, due to the high electronegativity of fluorine, fluorinated cations can introduce strong ionic and intermolecular bonding within the perovskite material. They are also helpful for passivating surfaces, thereby suppressing their chemical reactivity in the presence of moisture or contact with water [34]. The lattice distortions induced by fluorinated cations, which are usually even larger than MA, result in a broad and robust photoluminescence spectrum upon excitations of samples with light sources at frequencies below the bandgap [35]. Furthermore, recent theoretical exploration suggested that partial substitution of MA with Fluorinated cations induces not only enhanced intrinsic stability but could also boost the device stability under applied bias during operation due to the suppression of ionic migration [24,35,36].

Fluorinated organic cation’s partial or total substitution is increasingly being reported as an efficient way to tune the properties of the perovskite and enhance their stability [34,37] by strengthening some of the weak hydrogen bonds between the organic cations and the surrounding [PbI_3_] framework [36].

Nevertheless, it is important to find the critical balance between the ionic size and concentrations of large cation incorporation. The larger cations in a mixed-cation perovskite can suppress the hysteresis and aging of the device while reducing the bandgap [38]. Besides affecting the size of the unit cell, the organic cation also helps to exert a screening effect of the moving ionic charges [39]. The Pb-I-Pb angles are severely affected by the strong steric and electrostatic interactions between the organic cation and the Pb-I lattice [29]. Furthermore, Garoufalis et al. pointed out the importance of the orientational disorder of the molecular cation by looking into their impact on the perovskite nanoparticles, as the orientational disorder influences the structural and electronic properties [40,41]. Though the fine-tuning of the parent perovskite stands as a never-ending endeavor, a thorough study of their structure–property relation seems to be missing in majority of the reports available.

In our previous reports, the data-driven and machine-learning-aided investigations we discussed the perovskites’ structural deformation induced by the various mixed-cation compositions of MA_1−x_*A*_x_PbI_3_, where *A* is the additive cation [42,43,44,45,46]. We reported that upon mixing *A*-site cations in MAPbI_3_ perovskite framework, cations with a larger size than MA behave differently in stabilizing the framework than smaller cations. This suggests that different cations exert on the [PbI_3_] inorganic network a variety of cage distortion and stress mechanisms. In this work, we ought to deepen this understanding when large cations are incorporated in the perovskites by comparing the behaviors of ehtylammonium (EA; [CH_3_CH_2_NH_3_]^+^) and 2-fluoroethylammonium (FEA; [CH_2_FCH_2_NH_3_]^+^) cations and then search for interesting compositions that help minimize the enthalpy of mixing. This is achieved by employing manual DFT calculations and automated workflows guided by Bayesian optimization (BO) to explore the energy landscape of these mixed-cation perovskites at different concentrations and molecular orientations, taking enthalpic effects into account.

## 2. Results and Discussion

### 2.1. Ionic Size

A hybrid organic–inorganic perovskite, *A*PbI_3_, is composed of a positively charged molecular cation ($A^{+}$) accommodated within a [PbI_3_]^−^ network where the molecule plays a role in charge balance. The molecular size, shape, and dipole moment determine the perovskite’s intrinsic stability as well as the three dimensional (3D) connectivity of the PbI_6_ octahedral network via the electrostatic interaction as well as non-covalent interactions. As illustrated in Figure 1a, changes in the molecular occupancy of the cavity benefit from the flexibility of the PbI_6_ octahedra and their ability to elongate, tilt and rotate hence allowing the preservation of the 3D connected network, which is desirable for optoelectronic and catalytic applications.

Figure 1b–d displays Electrostatic potential (ESP) maps of the MA, EA, and FEA organic cations considered in the present work. ESP analysis is a useful metric for analyzing the electrostatic properties of a given cation and its suitability to occupy the cavity within the perovskite framework. The ESP maps reveal that all three singly positive charged molecular cations are electropositive around all the constituent atoms.

As a common feature in all three cations, the most electron-deficient sites are located within the amine (−NH_3_) groups. Thus, once any of the three molecules are hosted within the perovskite structure cavity, the electron-deficient amine groups would exert the strongest ionic interactions on the surrounding [PbI_3_]^−^ network. Focusing on the −CH_3_ groups of the three molecules, one might note that the electropositivity is significantly different around the terminal C atoms. EA and FEA molecules that have larger radii and stronger dipole moments than MA (see Table 1) have a less positive −CH_3_ and −CH_2_F terminal group as compared to the −CH_3_ group in MA. This feature suggests that the electrostatic attraction of the terminal C group with the [PbI_3_]^−^ network is weaker in the case of the larger molecules like EA and FEA compared to MA. However, this ESP analysis does not account for non-covalent interactions, such as hydrogen and halogen bonding, that are known to play an essential role in stabilizing the hybrid perovskites structure [36].

The complete substitution of MA with significantly larger cations EA and FEA, to form EAPbI_3_ and FEAPbI_3_, led to a Goldschmidt tolerance factor (τ) exceeding 1 (see Table 1), which is indicative that the 2D hexagonal phase might be favored over the highly symmetric 3D cubic phase.

On the other hand, the partial substitution of MA with a larger cation at different concentrations brings several advantages for the stability and efficiency of the devices [34,47] while preserving the 3D connectivity of the octahedral network [46]. For example, the large dipole moment of molecular cation in mixed-cation halide perovskites [46] is a desired property that is known to enhance the charge carrier separation when the incident light generates photoexcited electron-hole pairs. In addition, the alignment of these molecules was found to be favorable for the hole transfer at the interfaces of the perovskite light absorbers [6,48].

The empirical cubic perovskite stability indicators, such as the tolerance factor and its enhanced versions, have been recently challenged and questioned after the successful synthesis of mixed-cation halide perovskites ($A_{1-x}A_{x}^{\prime }{\mathrm{PbI}}_{3}$) that crystallize in the 3D pseudocubic perovskite structures. Our group recently demonstrated that the enthalpy of mixing plays a significant role in enhancing the intrinsic stability of mixed-cation halide perovskites compared to the conventional single cation MAPbI_3_ material [46]. The enthalpy of mixing *A*PbI_3_ and A′PbI_3_ ($\Delta H_{\mathrm{mix}}$) is calculated according to:
(1)$$\begin{array}{c} \Delta H_{\mathrm{mix}}\left(A_{1-x}A_{x}^{\prime }{\mathrm{PbI}}_{3}\right)\\ =\Delta H\left(A_{1-x}A_{x}^{\prime }{\mathrm{PbI}}_{3}\right)-\left\{\left(1-x\right)\Delta H\left(A_{1-x}{\mathrm{PbI}}_{3}\right)+x\Delta H\left(A_{x}^{\prime }{\mathrm{PbI}}_{3}\right)\right\}, \end{array}$$
where ΔH is the enthalpy of formation for a given chemical formula of perovskite obtained from DFT calculations.

Although empirical models indicate that τ>1 due to the large ionic size of EA and FEA, it remains remarkably relevant to search for promising compositions by exploring the energy landscape of these mixed-cation perovskites at different cation mixture concentrations and orientations taking enthalpic and entropic effects into account.

### 2.2. Enthalpy of Cation Mixing

Here, we report the enthalpies of mixing, $\Delta H_{\mathrm{mix}}$, upon the substitution of MA with EA and with FEA within a perovskite framework, as shown in Figure 2. The resulting mixed-cation perovskites have the chemical formula ${\mathrm{MA}}_{1-x_{\mathrm{EA}}}{\mathrm{EA}}_{x_{\mathrm{EA}}}{\mathrm{PbI}}_{3}$ and ${\mathrm{MA}}_{1-x_{\mathrm{FEA}}}{\mathrm{FEA}}_{x_{\mathrm{FEA}}}{\mathrm{PbI}}_{3}$, where $0\leq x_{\mathrm{EA}}\leq 1$ and $0\leq x_{\mathrm{FEA}}\leq 1$, respectively. Among the fully optimized supercells, few non-perovskite structures have emerged where the connectivity between and within the [PbI_6_] octahedra was broken. One can note that, across the full compositional range, the non-perovskites represent the minority of three and five structures among 46 relaxed supercells of ${\mathrm{MA}}_{1-x_{\mathrm{EA}}}{\mathrm{EA}}_{x_{\mathrm{EA}}}{\mathrm{PbI}}_{3}$ and 61 of ${\mathrm{MA}}_{1-x_{\mathrm{FEA}}}{\mathrm{FEA}}_{x_{\mathrm{FEA}}}{\mathrm{PbI}}_{3}$, respectively. Thus, in the remainder of the article, we exclude these non-perovskites from the dataset and investigate only the perovskite structures.

Structurally, it is well accepted that the unequal molecular sizes typically induce some local distortion in the [PbI_6_] octahedral network that often results in a positive value of $\Delta H_{\mathrm{mix}}$. Interestingly, in the case of EA and FEA, one might notice a negative $\Delta H_{\mathrm{mix}}$ region as marked by the blue shaded region, where the extensive sampling of initial molecular orientations led to the exploration of the energy landscape.

Cation segregation is a common problem often encountered during the substitution of MA with other cations, such as Cs, K, Rb, FA, and GUA. Cation mixture relaxes the lattice contraction, thereby balancing the lattice strain and homogenizing alloyed organic–inorganic perovskites [49,50,51]. Our discovery of the perovskite structures representing minima at which $\Delta H_{\mathrm{mix}}<0$ meV/ion implies the relaxation of the pure perovskites by the cation mixing. Subsequently, the negative value of $\Delta H_{\mathrm{mix}}$ suggests a suppressed EA and FEA cation segregation within the perovskite structures at the corresponding concentrations.

Contrary to the empirical structural model predictions, most mixed-cation compositions, ${\mathrm{MA}}_{1-x_{\mathrm{EA}}}{\mathrm{EA}}_{x_{\mathrm{EA}}}{\mathrm{PbI}}_{3}$ and ${\mathrm{MA}}_{1-x_{\mathrm{FEA}}}{\mathrm{FEA}}_{x_{\mathrm{FEA}}}{\mathrm{PbI}}_{3}$, lead to the formation of stabilized 3D perovskite structures, despite the considerable molecular size difference between MA and its substituents, namely EA and FEA. Among the computed FEA-MA mixed perovskites compositions, the chemical formula MA_0.875_FEA_0.125_PbI_3_ turns out to have the lowest $\Delta H_{\mathrm{mix}}$ of −4 meV/ion. This negative value implies that the perovskite [PbI_3_] network would be favorable to accommodating this given concentration of FEA molecules, despite their bulky size of 338 pm. It is worth noting that the optimal concentration of the large cation replacement of MA should increase the intrinsic stability without compromising the desired structural and electronic properties. Consequently, extensive analysis and discussion of the bandgap and structural properties of the mixed cations are provided in the Appendix A.

Figure 3 displays the distribution of the $\Delta H_{\mathrm{mix}}$ computed data as a function of the concentration of the substituted molecules EA and FEA, represented by using the box-and-whisker plots (boxplots). It can be noticed that for EA incorporation, most $\Delta H_{\mathrm{mix}}$ values lay within an interquartile range of less than 5 meV/ion at most $x_{\mathrm{EA}}$ values. Interestingly, for $\Delta H_{\mathrm{mix}}$ calculated for FEA incorporation, most $\Delta H_{\mathrm{mix}}$ values lay within an interquartile range of less than 10 meV/ion at most $x_{\mathrm{FEA}}$ values except at $x_{\mathrm{FEA}}=1$, which spreads the most widely.

### 2.3. Cation Alignment

Figure 4a illustrates the MA alignment distribution in the entire dataset of computed perovskite crystal structures within a 48-fold symmetry group (Oh). In the perovskites, MA cations prefer to align in parallel to the diagonal direction (〈111〉; θ=4/π and ϕ=4/π). This preferential orientation originates from the tendency to form hydrogen bonds between MA and the I^−^ anions. Meanwhile, the ionic radii of both EA and FEA cations exceed that of MA, hence stretching the cavity of the [PbI_3_] network. As shown in Figure 4b,c for EA and FEA, respectively, these cations align toward the faces of the cube (θ=0, ϕ=0) by avoiding the short-range repulsion between cation and I^−^. EA and FEA have a preference to align further toward the faces mainly because of their larger ionic radius if compared to MA. This cation size effect results in the average direction deviating from the diagonal direction as the EA and FEA concentrations increase (Figure 4d). The cations’ direction subsequently influences the formation energies by deforming the [PbI_3_] network and affecting the non-covalent interactions, such as the hydrogen bond and the halogen–halogen bond.

Hence, the structural stability of fully substituted FEAPbI_3_ perovskite crystal structures appears to be strongly dependent on the FEA molecular alignments. This impact is much stronger than the case with MA cations in MAPbI_3_ [53] which is due to the larger molecular size and dipole moment.

Phase segregation into either a 3D/2D or a 3D/1D mixture [31,47,54], when τ>1, implies that large-cation incorporation affects the restricted region while retaining the individual features within the inorganic network. Furthermore, empirically, a pure perovskite compound turns out a hexagonal phase favorably when τ>1. As we consider the enthalpy change within the cubic phase, the above stability estimation by mixing cations lacks the phase transition assessment. However, the comparisons of varying molecular alignments demonstrate its importance within the organic network, stabilizing the compound.

The search of DFT calculated structures giving the minimal enthalpy of mixing for a given concentration of substituted large molecular cations requires extensive sampling of randomized initial orientations within the perovskite supercell, making energy landscape exploration challenging. Detailed analysis of the current dataset points towards some promising large cation concentrations and trends in the cation alignments that would enable a guided search for the optimal concentration of large cation replacement of MA, without compromising the desired structural and electronic properties. The outcomes of the above analysis could be utilized by researchers to manually focus their compositional search on a narrower set of concentrations and molecular alignments. However, this approach might introduce bias into the search space. Furthermore, relevant concentrations could be missed as it is highly dependent on the researcher’s intuition and background. The following section reports our findings from Bayesian optimization (BO) to guide the optimal molecular concentrations and alignments.

### 2.4. The Search for Optimal Concentration of Large Cation Replacement of MA: Bayesian Optimization Aided Design

Although DFT is a compelling method for investigating materials’ structural and electronic properties, its principal limitation is the computational expense incurred for extensively sampling all possible combinations of cation concentrations and orientations within a large supercell in a high-throughput scheme.

By increasing the supercell size, it is burdensome to empirically identify the minima of $\Delta H_{\mathrm{mix}}$ with confidence because not only does the DFT computation cost grow, but also the count of mixed-cation combinations to compute enlarges dramatically. For example, consider MA_0.5_FEA_0.5_PbI_3_ within 2 × 2 × 2 supercell. Without any symmetry, the number of mixed-cation initial configurations is ^8^*C*_4_ = 70. On the other hand, for a 3 × 3 × 3 supercell, the number of possible initial configurations reaches ^27^*C*_13_ (and ^27^*C*_14_) = 20,058,300, making their exploration using DFT calculations prohibitively expensive.

We explore the applicability and efficiency of an automated workflow based on Bayesian Optimization (BO) in finding the structures that have minimal enthalpy of mixing, taking into account the large cation concentration and the cation alignment. Our initial dataset of mixed large-cations was constructed using computationally expensive DFT calculations for the function $\Delta H_{\mathrm{mix}}$. A manual inspection of the dataset suggests that there are regions in the two-dimensional search X space ( *molecular concentration* and *alignment*) where $\Delta H_{\mathrm{mix}}$ achieves low values pointing toward the minimum. These are regions where the optimal *concentration* might exist and where large cation replacement of the MA cation can be carried out without compromising the desired structural and electronic properties of the resulting perovskite. However, instead of proceeding manually, we can use BO to guide and automate our search towards the optimal configuration, of which $\Delta H_{\mathrm{mix}}$ is the lowest, in a computationally efficient manner. BO is ideally suited for this task because (i) each data point is obtained using a computationally expensive DFT calculation, and (ii) we do not have access to the underlying $\Delta H_{\mathrm{mix}}$ function but only its values at the expensively computed points. BO can incrementally guide our search towards the minimum of $\Delta H_{\mathrm{mix}}$ despite the absence of the gradient in the observation by developing a computationally inexpensive surrogate function to suggest the following configuration in the search space for which the DFT calculation should be invoked.

In our particular scenario, the search for the minimum is carried out within the 2D-space domain X composed of (concentration, alignment) while the function *f* we aim to optimize is $\Delta H_{\mathrm{mix}}$ where each evaluation requires an expensive DFT computation. Full BO computational details are given in the Methodology Section.

Figure 5 presents the exploration and exploitation of 100 evaluations of $arg\min_{(x,S)\in D}\Delta H_{\mathrm{mix}}^{\mathrm{BO}}\left(x,S\right)$ using Bayesian optimization (BO) with the posterior mean (μ(x), see Section 3.2), applied on the DFT-calculated dataset (D). The BO searched for the lowest $\Delta H_{\mathrm{mix}}$ values as a function of the concentration and molecular alignment. Comparing both panels in Figure 5, we observe that the MA/EA perovskite points are clustered in different *S* regions to the MA/FEA perovskite ones, whereas both MA/EA and MA/FEA data points are clustered within the concentration domain, 0<x<0.5. Notably, the cluster of the searched points is located at the points where $\Delta H_{\mathrm{mix}}$ values are relatively low, while the BO model searches for the global minimum with *x* and *S*. The rest of the region is explored sparsely as BO determines that the probability of finding the minima there is low. This is a manifestation of the exploration–exploitation trade-off. BO will always have a non-zero probability of exploring regions of high uncertainty with the possibility of finding a minimum in those regions and this property allows BO to escape settling for local minima.

### 2.5. DFT Workflow Automation Aided by Bayesian Optimization

In the above DFT-calculation sections, we had manually selected the mixed-cation concentration when preparing the initial structures of DFT input in a sequential decision way. The number of initial configurations also limited us to conducting DFT calculations at each concentration. In this section, a BO model implemented in the workflow plays the role of the researcher who prepares the initial structures balancing exploration and exploitation.

As shown in Figure 6, we present the results of our automated BO-guided DFT workflow that searches for the minimum $\Delta H_{\mathrm{mix}}^{\mathrm{BO}}-\mathrm{DFT}$ across the cations’ concentrations in MA_1−x_EA_x_PbI_3_ and MA_1−x_FEA_x_PbI_3_ actively and autonomously. The Rocketsled [55] framework has been adopted to build our automated workflow that searches for the minimum $\Delta H_{\mathrm{mix}}^{\mathrm{BO}}-\mathrm{DFT}$ by exploiting and exploring the 2D space domain of large cation concentrations and molecular alignments.

As we consider the molecular orientation, the BO model efficiently explored the search space to identify an additional minimum for $\Delta H_{\mathrm{mix}}^{\mathrm{BO}}-\mathrm{DFT}$ compared to the manual procedure. A comparison of Figure 6 with Figure 2 shows that, using BO, we were able to obtain a *lower* minimum $\Delta H_{\mathrm{mix}}^{\mathrm{BO}}-\mathrm{DFT}$ value of −20 meV compared to −4 meV from the manually-prepared DFT calculation. This enhanced energy landscape exploration is mainly attributed to extensive molecular orientation sampling carried out automatically by the BO procedure.

In the manually-prepared DFT results of the MA_1−x_EA_x_PbI_3_ perovskites, where the initial molecular orientations were given randomly, the minimum was located at x=0.250. Interestingly, the BO-guided automated search workflow located another lower minimum of $\Delta H_{\mathrm{mix}}^{\mathrm{BO}}-\mathrm{DFT}$ at around x=0.375 by exploiting and exploring molecular orientations during the preparation of input perovskite structures.

Meanwhile, MA_1−x_FEA_x_PbI_3_ perovkskites show the minimum $\Delta H_{\mathrm{mix}}^{\mathrm{BO}}-\mathrm{DFT}$ at the x=0.724; which corresponds to x=0.725 in the (2 × 2 × 2)-supercell perovskites. Although the BO model also exploited the lower concentration at around x=0.250, it also explored the higher concentration. As a result of the BO method, we found that the global minimum is around x=0.725. It was unlikely that this *global* minimum could be located using manual inspection as it is not consistent with the common belief of researchers that large cations do not fit well within the perovskite structure.

Subsequently, we have extended the use of automated DFT workflows aided by Bayesian optimization to explore other interesting concentrations of MA_1−x_FEA_x_PbI_3_ 3 × 3 × 3 supercells. As shown in Figure 7a, the BO model was able to locate a potential minimum of $\Delta H_{\mathrm{mix}}^{\mathrm{BO}}-\mathrm{DFT}$ with a large cation concentration and molecular alignment. We obtain the minimum point at x=0.727 similarly to the outcomes of the BO-guided DFT workflow in 2 × 2 × 2 supercells. The only distinction is that the minimum values are quantitatively higher across the FEA concentration ($x_{\mathrm{FEA}}$) in 3 × 3 × 3 supercells than those in 2 × 2 × 2 supercells mainly because the high degrees of freedom tend to lower the total energy of pure perovskites’ structures. Figure 7b shows the convergence of the automated BO-guided DFT workflow searches in finding a potential global minimum as a function of the number of BO evaluations. The BO optimizer was able to locate the minimum mixing enthalpy around the 23rd iteration; that minimum continues to sustain until the 65th iteration.

Hence, we have demonstrated the efficacy of DFT workflow automation aided by Bayesian optimization to accelerate the compositional search.

## 3. Computational Methods

### 3.1. Dft Calculation Methods

We performed the DFT calculations of the (2 × 2 × 2)-supercell perovskites, using the Vienna Ab-initio Simulations Package (VASP) [56,57,58]. The projector-augmented wave [59,60] method described the wavefunction of the valence electrons at the Perdew–Burke–Ernzerhof (PBE) [61,62] functional level. The plane–wave cutoff was set at 520 eV for all the calculations. Besides, to make up for the GGA’s inadequate dispersion interactions, we include the Tkatchenko–Scheffler van der Waals correction [63]. We optimized the structures until the energies and forces were converged within 10^−7^ eV/atom and 0.01 eV/Å, respectively. Moreover, the calculated energies of the structures were obtained with the Monkhrost–Pack scheme [64] 4 × 4 × 4 k-point grid.

While we varied the content of the organic cations in a perovskite framework, we placed methylammonium (MA), ethylammonium (EA), and 2-fluoroethylammonium cations (FEA) in the vacant organic-cation sites. Their locations were undefined when we sampled the mixed-cation perovskites. Accordingly, in an ${\mathrm{MA}}_{1-x}A_{x}{\mathrm{PbI}}_{3}$ perovskite (where *A* = EA and FEA; and 0<x<1), the MA and *A* cations were distributed in random locations among the *A*-cation sites, in the cavity of the [PbI_3_]^−^ network. Besides, the orientation of each cation was rotated differently in the initial coordinates. To manage the workflow, we customized FireWorks [65], and Pymatgen [66] tools were used when we set the perovskites and analyzed the optimized coordinates. Accordingly, we optimized and sampled 46 ${\mathrm{MA}}_{1-x_{\mathrm{EA}}}{\mathrm{EA}}_{x_{\mathrm{EA}}}{\mathrm{PbI}}_{3}$ and 61 ${\mathrm{MA}}_{1-x_{\mathrm{FEA}}}{\mathrm{FEA}}_{x_{\mathrm{FEA}}}{\mathrm{PbI}}_{3}$ compounds, where $0\leq x_{\mathrm{EA}}\leq 1$ and $0\leq x_{\mathrm{FEA}}\leq 1$, respectively.

When we calculated the cations’ dipole moment and electrostatic potential (ESP) in the gas phase, GAMESS 2020 R2 [67] was performed with B3LYP/6-311+G(d,p). Using these optimized coordinates, we obtained the effective ionic radii of the organic cations according to the method suggested by Kieslich et al. [68].

### 3.2. Bayesian Optimization Methodology

Bayesian Optimization (BO) addresses the problem of finding a global optimization of a function whose form is not available but is accessible through (possibly noisy) function evaluation [69]. Thus, if f:X→Y, which needs to be optimized, then *f* is accessible through a set of tuples $\{{\left(x_{i},f\left(x_{i}\right)\right\}}_{i=1}^{n}$. Furthermore, it is assumed that function calls to *f* are computationally expensive, and thus a small number of evaluations should be carried out to arrive at the optimal.

In BO, one begins with a prior distribution on the functional forms of *f* and, after each evaluation, the prior is updated using the Bayes rule to arrive at a posterior distribution. The class of prior distributions is called the surrogate model, which is assumed to close under the application of a Bayesian update: if F is the surrogate model and g∈F, then after the function evaluation, the new function $g^{\prime }\in F$. Since the assumption is that the evaluation of the (unknown) function is computationally expensive, the selection of *x*, the next evaluation element, is guided by an acquisition function, *a*, which is derived from the latest posterior function $g^{\prime }$ and is relatively (computationally) cheaper to evaluate. After *a* proposes the next point to evaluate, the function call *f* is made and that in turn updates the posterior distribution on F. This process is repeated until a computational budget is exhausted or $a^{\prime }$ proposes an evaluation point close to the one that has already been computed. A key characteristic of the acquisition function *a* is that it is designed to trade-off exploration vs exploitation. Exploration is designed to propose evaluation points for which the model has high uncertainty about the function value of *f*, while exploitation does the converse, that is, suggests points where there is a higher certainty about the value of *f*.

The most common surrogate model in BO is a Gaussian Processes (GP), which is a generalization of a multivariate Gaussian (Normal) distribution to functions [70]. A GP is defined by a mean function μ(x) and a covariance function *k* and is denoted as GP(μ(x),k). Problem specific information and constraints can be incorporated in *k*. A common covariance function is the Gaussian kernel $k\left(x,y\right)=\exp\left(-\frac{{∥x-y∥}^{2}}{h}\right)$, where ${∥x-y∥}^{2}$ is the Euclidean norm and *h* is a length-scale parameter. Generalizations of the Gaussian kernel, including the Matern kernel and anisotropic kernels, have been used.

For the acquisition function *a*, several choices are available. A common choice is the *Expected Improvement*. Suppose $x^{*}$ is the minimal value of *f* observed so far. Then define a function *a* as:
$$a\left(x\right)=E_{f}(\max\left(0,x^{*}-f\left(x\right)\right).$$

The next evaluation point is then $\arg\max_{x}a\left(x\right)$, that is, a point is selected where the expected improvement is maximized.

Our constructed BO models guide the cations’ concentrations and their initial coordinates in the DFT workflow automation. When we set the search domains, in addition to concentration variable *x*, we took the $\Delta a_{\max}$ as the feature domain to prepare the molecular alignments in the DFT input. Because the molecular alignment relaxes during the ionic optimization calculations, the BO model sets the initial molecular alignments around the 〈111〉 direction instead of the relaxed structures’ orientation order parameter (*S*). Here, we put $\Delta a_{\max}$, the maximum angle between the 〈111〉 direction and molecular direction, to launch the DFT geometry relaxation tasks. Then, the workflow relocates the cations by allowing them in this molecular orientation range within the 48-fold symmetry group, preparing the input coordinates. For example, when $\Delta a_{\max}=0$, all the cations are aligned in the 〈111〉 direction in the perovskite. Besides, when $\Delta a_{\max}=\pi /8$, the cations are oriented randomly. Subsequently, the DFT relaxation task returns the estimated $\Delta H_{\mathrm{mix}}^{\mathrm{BO}}-\mathrm{DFT}$ once the DFT geometry relaxation is completed. The input preparation of initial structures to be explored as a function of input with the concentrations and molecular alignments of the organic cations (*x* and $\Delta a_{\max}$) in 2 × 2 × 2 and 3 × 3 × 3 supercells. The expected improvement (EI) acquisition function enabled us to sample the points to be explored and exploited. The $\Delta H_{\mathrm{mix}}$ values are predicted by the Gaussian Process (GP) regressor. Meanwhile, when we carried out the BO-guided workflow automation to assess $\Delta H_{\mathrm{mix}}^{\mathrm{BO}}-\mathrm{DFT}$, the workflow sets up a 2 × 2 × 2-grid mesh of the **k**-points for the 2 × 2 × 2 supercell and a gamma-centered 1 × 1 × 1-grid mesh for the 3 × 3 × 3 supercell during the VASP calculations.

## 4. Conclusions

Large molecular cation incorporation in halide perovskites has recently emerged as an appealing strategy for enhancing materials and device stability. We evaluate the impact of large-sized cations’ incorporation within the perovskite framework by mixing MA with two representative cations, EA and FEA. The analysis of the DFT calculations shows the importance of extensive sampling in the cation-concentration and orientation domains. Furthermore, the entropy-favored configurations hinder finding the global minimum. While the cation orientation order correlates with the local strain and instability, it suggests that orderly oriented cations can incorporate large cations at a given higher concentration.

Allowing for the effects at different concentrations and molecular orientations, our Bayesian optimization (BO)-guided workflow automation explored the energy landscape of these mixed-cation perovskites. The BO-guided workflow efficiently identifies the minimum enthalpy of mixing compared to the manual procedure. The enhanced energy landscape exploration is mainly attributed to extensive molecular orientation sampling aiming at the global minimum. This approach has the potential to reduce the DFT computational expense associated with searching for optimal compositions and might lead to the establishment of design principles for large molecular cation incorporation in halide perovskites expandable to other classes of materials.

## Figures and Tables

**Figure 1 molecules-26-06184-f001:**
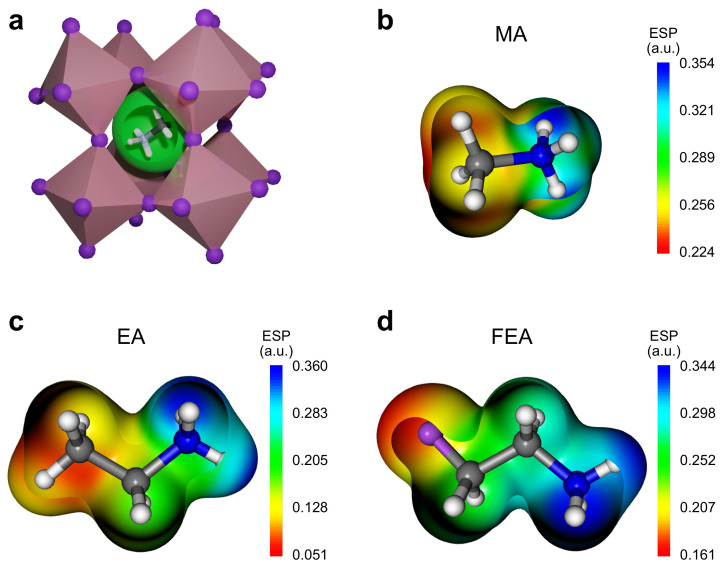
(**a**) Crystal structure of a *A*PbI_3_ perovskite. An organic cation is surrounded by the connected PbI_6_ octahedra and is accommodated within the cavity shown as green sphere. Electrostatic potential (ESP) maps of organic cations: (**b**) methylammonium, MA; (**c**) ethylammonium, EA; and (**d**) 2-fluoroethylammonium, FEA. The ESP potential is mapped in Hartree atomic units (a.u.) for a density isosurface of *ρ* =0.01 e·Bohr^−3^.

**Figure 2 molecules-26-06184-f002:**
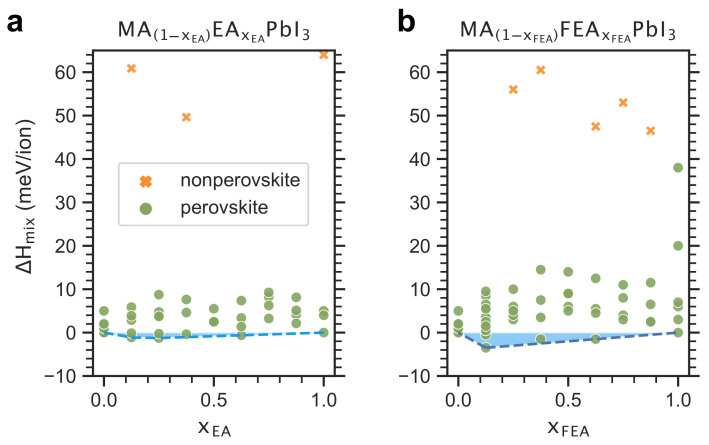
Enthalpies of mixing, $\Delta H_{\mathrm{mix}}$ (meV/ion), as a function of the large molecule concentration $x_{\mathrm{EA}}$ and $x_{\mathrm{FEA}}$ for (**a**) ${\mathrm{MA}}_{1-x_{\mathrm{EA}}}{\mathrm{EA}}_{x_{\mathrm{EA}}}{\mathrm{PbI}}_{3}$ and (**b**) ${\mathrm{MA}}_{1-x_{\mathrm{FEA}}}{\mathrm{FEA}}_{x_{\mathrm{FEA}}}{\mathrm{PbI}}_{3}$. The orange crosses and green circles distinguish between non-perovksite and perovskite structure, respectively. The blue shaded areas denote the resulting region of enhanced stability with respect to the parent $\mathrm{MA}{\mathrm{PbI}}_{3}$, where $\Delta H_{\mathrm{mix}}<0$ meV/ion.

**Figure 3 molecules-26-06184-f003:**
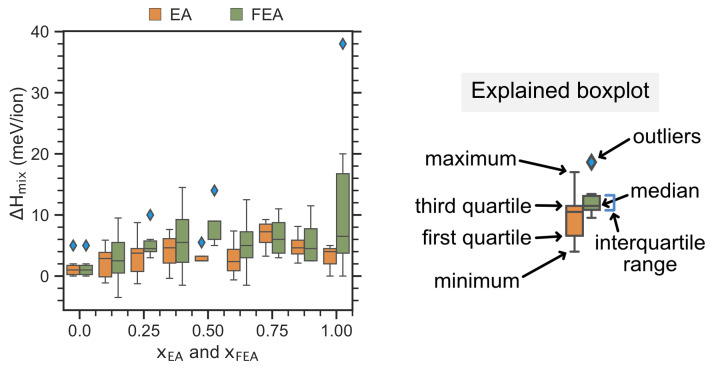
$\Delta H_{\mathrm{mix}}$ distribution in the dataset as a function of the concentration of the substituted molecule. We use Seaborn’s boxplot [52] to display the properties and their distribution. The outliers are marked by the blue diamonds. The different parts of a boxplot are explained in the right panel showing the five-number summary of a set of data: minimum, lower quartile, median, upper quartile, and maximum.

**Figure 4 molecules-26-06184-f004:**
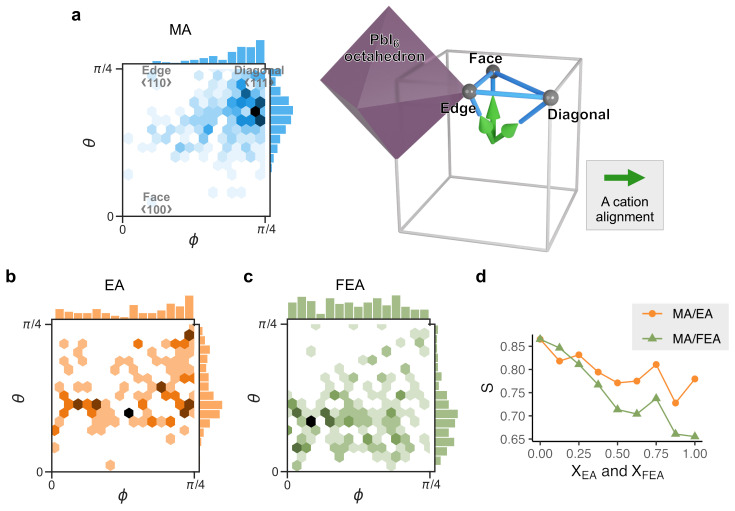
Density map of molecular alignment vectors projected on spherical coordinates represented within 48-fold symmetry group (Oh) for (**a**) MA with the illustration of the representative directions, (**b**) EA, and (**c**) FEA. Here, the alignment vectors of a cation are constructed by connecting the terminal C, N, F atoms. The darker color indicates the highest incidence as also shown from the histograms constructed by binning the spherical angles. Face, edge and diagonal refer to the alignment of the molecules with respect to the perovskite cage, and correspond to the 〈100〉 (θ=0 and ϕ=0), 〈110〉 (θ=π/4 and ϕ=0) and 〈111〉 (θ=π/4 and ϕ=π/4) directions. (**d**) Orientation order parameter $S=\frac{3}{2}\langle \cos^{2}\alpha \rangle -\frac{1}{2}$ as a function of the substituted large cation concentration, where α is the angle between 〈111〉 and the molecular directions.

**Figure 5 molecules-26-06184-f005:**
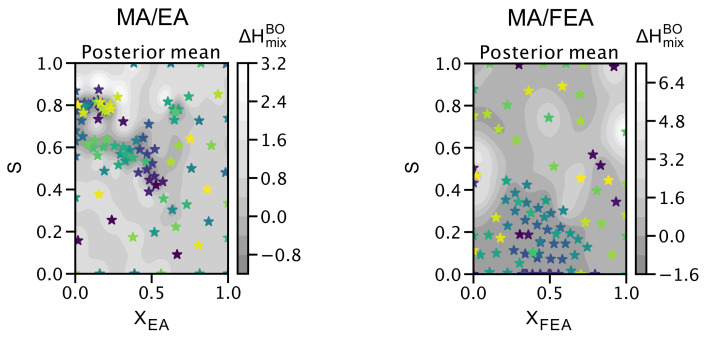
Exploration and exploitation of 100 evaluations of $arg\;\min_{(x,S)\in D}\Delta H_{\mathrm{mix}}^{\mathrm{BO}}\left(x,S\right)$ (in meV/ion), where D denotes the domain from the dataset. As the 100 cycles of the BO evaluation proceed, the points are marked gradually in order of sampling on a color scale from dark blue to yellow stars.

**Figure 6 molecules-26-06184-f006:**
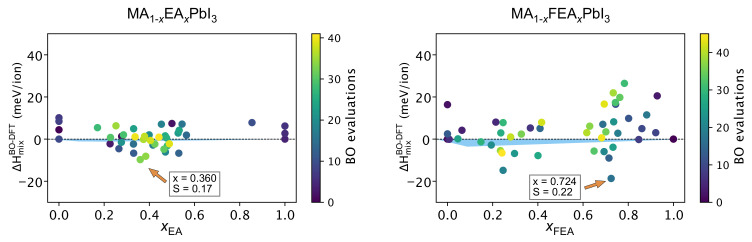
Exploration and exploitation of $arg\min\;\Delta H_{\mathrm{mix}}^{\mathrm{BO}}-\mathrm{DFT}$ (in meV/ion), as the post-analyzer, processes the data during the DFT workflow automation, where the next best cation’s concentration and orientation are selected by BO. As the BO search proceeds, the points are marked gradually in order of sampling on a color scale from dark blue to yellow circles. The configuration giving the minimum enthalpy found using the BO aided automated DFT workflow is pointed to by the orange arrow. The blue shaded areas denote the DFT results where $\Delta H_{\mathrm{mix}}<0$ meV/ion, conducted manually.

**Figure 7 molecules-26-06184-f007:**
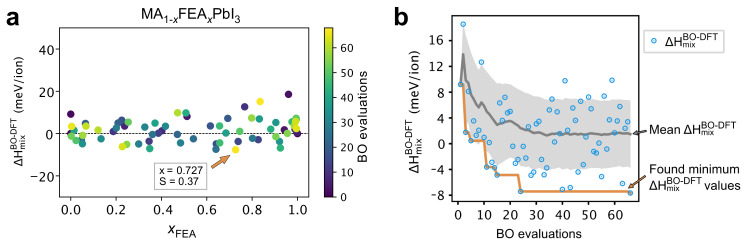
(**a**) Exploration and exploitation of $arg\min\;\Delta H_{\mathrm{mix}}^{\mathrm{BO}}-\mathrm{DFT}$ (in meV/ion) for 3 × 3 × 3 MA_1−x_FEA_x_PbI_3_ supercell, as the post-analyzer processes the data during the DFT workflow automation and the next best cation’s concentration and orientation are selected by BO. As the BO evaluation progresses, the points are marked gradually in order of sampling on a color scale from dark blue to yellow circles. The configuration giving the minimum enthalpy found using the BO aided automated DFT workflow is pointed to by the orange arrow. (**b**) The number of BO evaluations and found $\Delta H_{\mathrm{mix}}^{\mathrm{BO}}-\mathrm{DFT}$ values of the BO for the 3 × 3 × 3 supercell. The blue circles are the DFT calculated $\Delta H_{\mathrm{mix}}^{\mathrm{BO}}-\mathrm{DFT}$ as the BO evaluation progresses. The orange line delineates the found minimum at each BO interation $\Delta H_{\mathrm{mix}}^{\mathrm{BO}}-\mathrm{DFT}$ and the gray line illustrates the mean values.

**Table 1 molecules-26-06184-t001:** The effective ionic radii and dipole moment of the used organic cations. And the tolerance factor (TF) in *A*PbI_3_ is calculated according to $\tau =\left(r_{A}+r_{I}\right)/\left(\sqrt{2}\left(r_{\mathrm{Pb}}+r_{I}\right)\right)$, where $r_{i}$ is the ionic radius of the corresponding *i* ion.

Cation	Formula	Ionic Radius (pm)	Dipole Moment (Debye)	τ
MA	[CH_3_NH_3_]^+^	217	2.20	0.91
EA	[CH_3_CH_2_NH_3_]^+^	274	3.85	1.03
FEA	[CH_2_FCH_2_NH_3_]^+^	338	9.17	1.16

## Data Availability

The raw/processed data and recipe/workflow codes can be obtained by contacting the authors.

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
