# Peer review of "Design Principles of Large Cation Incorporation in Halide Perovskites"

_molecules, 2021, doi:10.3390/molecules26206184_

Round 1

Reviewer 1 Report

In this work, F. El-Mellouhi et al. have developed design principles for the incorporation of large cation in halide perovskite of the optimal stable compositions. Density functional theory calculations of the enthalpy of mixing was performed for large cations (ehtylammonium (EA) and 2-fluoroethylammonium (FEA)) incorporation. They have successfully explored the applicability of the Bayesian Optimization (BO) in finding the structures having a minimal enthalpy of mixing, also accounted for the large cation concentration and cation alignment. They have also validated an automated workflow designed to accelerate the compositional search, enabling researchers to cut down the computational cost and bias during the search for optimal compositions. The results presented in this work are interesting and provide new ideas for designing a stable mixed composition of perovskite. Therefore, I would like to recommend this manuscript for publication in “Molecules” after answering the given comments.

There are few comments on the manuscript:

  1. In the introduction, large cations discussion needs to elaborate with citing the other published papers like; Sustainable Energy Fuels, 2021,5, 4327-4335; ACS Appl. Energy Mater. 2021, 4, 3, 2751–2760.
  2. 149 & 151 lines – electron-decient? Crosscheck this term
  3. According to the tolerance factor > 1 for such a large cation led to the hexagonal crystal structure. Is it possible for pure PbI2:FEA (1:1) perovskite to form 1D phase? Authors need to comment on this direction too.
  4. It would be better if authors can compare results with experimental data from the literature for validation or generalize the discussion for other similar mixed 3D/2D or 3D/1D perovskite materials on the incorporation of large cations.

Author Response

Reviewer #1, comment #1

In the introduction, large cations discussion needs to elaborate with citing the other published papers like; Sustainable Energy Fuels, 2021,5, 4327-4335; ACS Appl. Energy Mater. 2021, 4, 3, 2751–2760.

Authors reply

We thank the reviewer for drawing our attention to this point. As the reviewer suggested the articles, we include the sulfonium-based cation incorporation in lead halide perovskites, in addition to the commonly known the ammonium-based cations. In lines 88-90, it reads:

“Furthermore, sulfonium-based cations, such as trimethylsulfonium and trimethylsulfoxonium, have shown the formation of lower dimensional perovskites resulting in the stability increase, as their ionic size is bulkier than MA [30,31]”

Reviewer #1, comment #2

149 & 151 lines – electron-decient? Crosscheck this term.

Authors reply

We thank the reviewer. The typographical errors are corrected in this revised version.

Decient -> Deficient

Reviewer #1, comment #3

According to the tolerance factor > 1 for such a large cation led to the hexagonal crystal structure. Is it possible for pure PbI2:FEA (1:1) perovskite to form 1D phase? Authors need to comment on this direction too.

Authors reply

We truly appreciate this very relevant comment. Indeed, when the tolerance factor > 1, a perovskite turns out to be a hexagonal crystal. From this point of view, we agree with the reviewer to comment about the hexagonal phase. However, we focus on the cubic phase perovskites and their autonomous workflow, deploying a Bayesian Optimization model to enhance the computational workflow efficiency. Rather than looking into the phase stability between cubic and hexagonal phases, we explore the impact of A-cations concentration and their orientation order in this work. Furthermore, we emphasize the molecular alignment to stabilize the cubic phase. However, in this revised version, we remark the phase segregation of the mixed-cation perovskites, although we cannot prove the possibility of pure PbI2:FEA (1:1) perovskite seeing whether it forms either 1D or 3D phase. We add a paragraph as it reads in lines 255-262:

“Phase segregation into either 3D/2D or 3D/1D mixture [31,49,50], when τ > 1, implies that large-cation incorporation affects the restricted region while retaining the individual features within the inorganic network. Furthermore, empirically, a pure perovskite compound turns out a hexagonal phase favorably when τ > 1. As we consider the enthalpy change within the cubic phase, the above stability estimation by mixing cations lacks the phase transition assessment. However, the comparisons of varying molecular alignments demonstrate its importance within the organic network stabilizing the compound.”

Reviewer #1, comment #4

It would be better if authors can compare results with experimental data from the literature for validation or generalize the discussion for other similar mixed 3D/2D or 3D/1D perovskite materials on the incorporation of large cations.

Authors reply

We thank the reviewer for the note as the reviewer suggested that we generalize the discussion concerning the favorable phase when τ > 1. We noted in the introduction section in the manuscript that there are large-cation incorporation efforts. And, indeed, some reports revealed the phase segregation and showed mixed 3D/2D or 3D/1D perovskites. To compare the cubic phase stability with 1D and 2D perovskite, we need to carry out DFT calculations making a bigger supercell so that it requires the computing recourses much. This burdensome DFT calculation cost is one of the reasons why we propose the Bayesian-Optimization-guided DFT workflow. We will be able to present the direct comparison including other phases performing the Bayesian-Optimization-guided. It should be discussed in a separate work, following this research.

Reviewer 2 Report

In this manuscript entitled “Design Principles of Large Cation Incorporation in Halide Perovskites”, instead of using the broadly known tolerance factor, the authors applied enthalpy calculation to predict the stability of perovskite with large cation incorporated. This topic is interesting to the perovskite community. However, the manuscript still has some flaws, and I would therefore recommend this manuscript to be published in molecules after major revision. Below please find the comments.  

  1. The citations are missing in the manuscript. There are some references I would like to read, such as “(CH3)2NH2SnI3 is stable in water for 16 hours”. Does this refer to perovskite single crystal? Also, I would like to know the optoelectronic properties of the synthesized CH3FNH3PbBr as mentioned in the introduction.
  2. Page6 Figure2, MAPbI3 has been applied in solar cells for a long time. However, the calculated delta H is higher than 0. The authors need to explain this data, otherwise, the following calculations will not be justified.
  3. Page6 line 196, the authors need to explain why delta H < 0 implied suppressed segregation.

Author Response

Reviewer #2, comment #1

The citations are missing in the manuscript. There are some references I would like to read, such as “(CH3)2NH2SnI3 is stable in water for 16 hours”. Does this refer to perovskite single crystal? Also, I would like to know the optoelectronic properties of the synthesized CH3FNH3PbBr as mentioned in the introduction.

Authors reply

We thank the reviewer for drawing our attention to this point. In fact, the preceding sentences cited the references. As we correct the citation location at each compound introduction for (CH3)2NH2SnI3, this revised version helps the reader correlate the compounds with the references in lines 66-68. And Huang et al. reported CH3FNH3PbBr3 has an optical bandgap of 2.3 eV. We rephased the sentence mentioning the bandgap specifically, in line 98.

Reviewer #2, comment #2

Page6 Figure2, MAPbI3 has been applied in solar cells for a long time. However, the calculated delta H is higher than 0. The authors need to explain this data, otherwise, the following calculations will not be justified.

Authors reply

We understand the reviewer’s concern. We carried out multiple MAPbI3 compounds but started with different alignments for the molecular cations. And the ΔH values are referenced to the minimum values as we defined ΔH in the manuscript. Hence, while we have the MA cations of which directions are differently aligned in the perovskite structures, several MAPbI3 have upper enthalpy values than the minimum. Consequently, there are the values of ΔH > 0 in our results of MAPbI3. We discuss the well-aligned cations contribute to the lower values of ΔH and emphasize molecular alignment’s importance in calculating the stability in mixed-cation perovskites.

Reviewer #2, comment #3

Page6 line 196, the authors need to explain why delta H < 0 implied suppressed segregation.

Authors reply

We thank the reviewer for the constructive input. Minimizing the lattice strain contributes suppressing the segregation [Zheng, X. et al. ACS Energy Lett. 2016, 1, 1014–1020]. We revised the paragraph, as it reads, in lines 207-216;

“Cation segregation is a common problem often encountered during the substitution of MA with other cations, such as Cs, K, Rb, FA, and GUA. Cation mixture relaxes the lattice contraction, thereby balancing the lattice strain and homogenizing alloyed organic-inorganic perovskites [47-49]. Our discovery of the perovskite structures representing minima at which $\Delta H _\mathrm{mix} < 0$ meV/ion implies the relaxation of the pure perovskites by the cation mixing. Subsequently, the negative value of $\Delta H_\mathrm{mix} $ suggests a suppressed EA and FEA cation segregation within the perovskite structures at the corresponding concentrations.”

Reviewer 3 Report

Report on “Design Principles of Large Cation Incorporation in Halide Perovskites”
The authors have performed a thorough and well conducted theoretical study on mixed cation MA and
EA, FA halide perovskite structures.
The calculations are competently performed within the specific methodological framework and the
results are interesting. It is my opinion that the manuscript is suitable for publication after a few trivial
correction.
– Unfortunately, in the review version the citations were not properly shown, so I could not follow the
discussion which was closely related to specific literature references. Despite this technical difficulty
the existing bibliography looked adequate. However, I would like to draw the authors, attention to the
following articles Electron. Mater. 2021, 2(3), 382-393 and ACS Omega 2018, 3, 18917-18924
which are quite relevant to their work and could be included in their discussion.
-- In line 149 and 151 the term “electron-decient” should be changed to “electron-deficient”
-- In line 176 the term “it is remains” should be corrected to “it remains”
-- In the caption of Figure the units of ΔΗ is meV/ion. It is not clear what the “per ion” means (per
which ion?)
-- The density maps of Figure 4 is a nice and effective way of describing the orientation of the
molecular moieties. However, there is no such clear reference to other structural characteristics of the
the materials. I understand that there are too many structures to describe, but it would be nice if the
dominant trends could be presented in a more pictorial way (if possible).
-- In line 253 the term “computational cost” would be prefereable to “computational expense”.
-- In line 430 the “FA” should be corrected to “FEA”
-- It would be nice (if possible) to include some information about the electronic properties as well.
For example what happens to the band gap? Although it is not strongly dependent on the type and
orientation of the molecular moieties, perhaps it might be affected by the structural deformations that
these moieties induce.

The pdf has to be updated so that the referees can proceed with the reviewing process.

Anyway, I believe that the manuscript presents a well perfomed theoretical work, which deserves to be
published after these minor corrections.

Author Response

Reviewer #3, comment #1

Unfortunately, in the review version the citations were not properly shown, so I could not follow the discussion which was closely related to specific literature references. Despite this technical difficulty the existing bibliography looked adequate. However, I would like to draw the authors, attention to the following articles Electron. Mater. 2021, 2(3), 382-393 and ACS Omega 2018, 3, 18917-18924 which are quite relevant to their work and could be included in their discussion.

Authors reply

We truly appreciate this very relevant comment. The articles discuss the molecular cation orientation and the authors pointed out the importance of looking into the orientation order. In this revised version, we remark the impact of organic cations’ disorder on the structural and electronic features of perovskite at lines 121-124 by adding the recommended articles as Ref. [40] and [41].

Reviewer #3, comment #2

In line 149 and 151 the term “electron-decient” should be changed to “electron-deficient”.

Authors reply

We thank the reviewer. The typographical errors are corrected in this revised version.

Decient -> Deficient

Reviewer #3, comment #3

In line 176 the term “it is remains” should be corrected to “it remains”.

Authors reply

We thank the reviewer. The grammatical error is corrected in this revised version.

Reviewer #3, comment #4

In the caption of Figure the units of ΔΗ is meV/ion. It is not clear what the “per ion” means (per

which ion?).

Authors reply

We understand the reviewer’s concern. An APbI3 perovskite consists of 5 ions–a A+ ion, a Pb2+ ion, and three I- ions. Hence, we divide ΔΗ in energy per formula unit by 5.

Reviewer #3, comment #5

The density maps of Figure 4 is a nice and effective way of describing the orientation of the molecular moieties. However, there is no such clear reference to other structural characteristics of the  materials. I understand that there are too many structures to describe, but it would be nice if the dominant trends could be presented in a more pictorial way (if possible).

Authors reply

We thank the reviewer for suggesting a pictorial explanation in Figure4 (a). We added the illustration of the representative directions beside the structural characteristics map of MA. The added picture would help the readers to read the cations’ alignments.

Reviewer #3, comment #6

In line 253 the term “computational cost” would be prefereable to “computational expense”.

Authors reply

We thank the reviewer. As the reviewer suggested, we specified the sentence’s intention by changing the word.

Reviewer #3, comment #7

In line 430 the “FA” should be corrected to “FEA”.

Authors reply

We appreciate that the reviewer found the misspelled abbreviation in the conclusion section. Indeed, it should be “FEA”. We corrected in this revised version.

Reviewer #3, comment #8

It would be nice (if possible) to include some information about the electronic properties as well. For example what happens to the band gap? Although it is not strongly dependent on the type and orientation of the molecular moieties, perhaps it might be affected by the structural deformations that these moieties induce.

Authors reply

We thank the reviewer for this constructive suggestion. As a matter of fact, we calculated the bandgaps. However, as we focus on the Bayesian Optimization DFT workflow and its efficiency to search for the optimal mixture of two organic cation mixing, we set aside the bandgap discussion into the supporting information. Meanwhile, to guide the readers who are interested in the bandgap, we commented in lines 227-228, saying that the supporting information includes the discussion of the electronic and other structural deformation trend.

Round 2

Reviewer 2 Report

The authors have resolved my concerns and this manuscript is recommended to be published in the current version. 

Reviewer 3 Report

It is my belief, that the revised version of the manuscript meets the high standards of the journal and it is suitable for publication.